# Impact of Antimicrobial Stewardship Interventions on Appropriateness of Surgical Antibiotic Prophylaxis: How to Improve

**DOI:** 10.3390/antibiotics9040168

**Published:** 2020-04-09

**Authors:** Beatrice Tiri, Paolo Bruzzone, Giulia Priante, Emanuela Sensi, Monya Costantini, Carlo Vernelli, Lucia Assunta Martella, Marsilio Francucci, Paolo Andreani, Alessandro Mariottini, Andrea Capotorti, Vito D’Andrea, Daniela Francisci, Roberto Cirocchi, Stefano Cappanera

**Affiliations:** 1Antimicrobial Stewardship Unit, Department of medicine, St. Maria Hospital, 05100 Terni; Italy; s.cappanera@aospterni.it; 2Department of General and Specialist Surgery “Paride Stefanini”, 00100 Rome, Italy; paolo.bruzzone@uniroma1.it; 3Infectious Diseases Clinic, Department of medicine, St. Maria Hospital, 05100 Terni, Italy; giulia.priante1989@gmail.com (G.P.); c.vernelli@aospterni.it (C.V.); l.martella@aospterni.it (L.A.M.); 4Department of Critical Care Medicine and Anesthesiology, St. Maria Hospital, 05100 Terni, Italy; e.sensi@aospterni.it; 5Pharmacy Unit, St. Maria Hospital, 05100 Terni, Italy; m.costantini@aospterni.it; 6Department of General and Oncologic Surgery, St. Maria Hospital, 05100 Terni, Italy; m.francucci@aospterni.it; 7Hematology and Microbiology Laboratory, St. Maria Hospital, 05100 Terni, Italy; p.andreani@aospterni.it (P.A.); a.mariottini@aospterni.it (A.M.); 8Department of Mathematics and Informatics, University of Perugia, 06121 Perugia, Italy; andrea.capotorti@unipg.it; 9Department of Surgical Sciences, Sapienza University of Rome, 00100 Rome, Italy; vito.dandrea@uniroma1.it; 10Department of Medicine, University of Perugia, 06121 Perugia, Italy; d.francisci@unipg.it; 11Department of Surgical Sciences, University of Perugia, 06121 Perugia, Italy; Roberto.cirocchi@unipg.it

**Keywords:** surgical site infections, antimicrobial stewardship, surgical antibiotic prophylaxis, appropriateness, healthcare-associated infections, antibiotic resistance

## Abstract

Surgical site infections (SSIs) are the most common healthcare-associated infections. The appropriate use of Surgical Antibiotic Prophylaxis (SAP) is a key component to reduce SSIs, while its inappropriate application is a major cause of some emerging infections and selects for antibiotic resistance. We describe an Antimicrobial Stewardship (AMS) intervention on SAP appropriateness. The prospective study was conducted in an Italian hospital, in 12 main surgical units, and was organized in three subsequent phases, as follows. Phase 0: Definition of hospital evidence-based guidelines and a new workflow to optimize the process of ordering, dispensing, administering and documenting the SAP. Phase 1: We analysed 2059 elective surgical cases from January to June 2018 for three SAP parameters of appropriateness: indication, choice and dose. Phase 2: In July 2018, an audit was performed to analyse the results; we reviewed 1781 elective surgical procedures from July to December 2018 looking for the same three SAP appropriateness parameters. The comparative analysis between phases 1 and 2 demonstrated that the correct indication, the correct dose and the overall compliance significantly improved (*p*-value 0.00128, *p*-value < 2.2·10^16^ and *p*-value < 5.6·10^12^ respectively). Our prospective study demonstrates a model of successful antimicrobial stewardship intervention that improves appropriateness on SAP.

## 1. Introduction

Surgical site infections (SSIs) are now the most common (20%) and costly of all hospital-acquired infections. They represent a major clinical problem in terms of mortality, morbidity, length of stay and overall costs. SSIs are a global priority also because 20%–35% of them are caused by antibiotic-resistant strains [1,2].

The burden of infections in Europe is similar to the combined burden of three major infectious diseases (influenza, tuberculosis and HIV), and about 75% of the total burden of infections with antibiotic-resistant bacteria were associated with healthcare. Italy and Greece had a substantially higher estimated burden of antibiotic-resistant bacteria than other EU and EEA countries [3].

The appropriate usage of Surgical Antibiotic Prophylaxis (SAP) significantly reduces the risk of SSI [4,5], while the inappropriate usage increases SSIs, hospital costs and multidrug-resistant strains [6,7,8,9,10,11].

For this reason, international and national guidelines have been developed to guide clinicians in the optimal use of SAP [12,13,14].

Italy is one of the largest consumers of Broad-Spectrum Agents (BSAs) in Europe [15]. SAP contributes significantly to antibiotic consumption in Italy, as it accounts for 17.4% of all antibiotics prescribed in acute care hospitals [16]. However, the overall compliance with guidelines is poor and is associated with a high heterogeneity. Improvements in the appropriateness of SAP, defined as the correct indication, drug dose, route, timing of administration and duration are the major objectives of national health services. However, the means to achieve this goal have not yet been defined.

Antimicrobial Stewardship (AMS) is a systematic approach to improve the appropriateness of antimicrobial use in order to optimize the treatment of infections, to minimize the adverse effects associated with antibiotic use and to reduce antimicrobial resistance, toxicity and costs [17,18].

The 2016 definition by the Society for Healthcare Epidemiology of America (SHEA) describes AMS as a “set of coordinated strategies to improve the use of antimicrobial medications with the goal to enhance patient health outcomes, reduce antibiotic resistance, and decrease unnecessary costs”.

There is no universally recognized intervention for improving SAP appropriateness: we describe, in our prospective study, a successful AMS intervention on SAP appropriateness with a significant improvement in overall compliance to guidelines.

## 2. Materials and Methods

Our prospective study was conducted at “Santa Maria” hospital, which is a tertiary hospital in Terni, Umbria, Italy with 564 beds.

The AMS program at “Santa Maria” Hospital began in February 2016 as a multilevel program with a comprehensive approach to improving the optimization of antibiotic use that is based on international and national guidelines, but tailored to our hospital’s reality.

The AMS team has been in place from July 2018 and is a multidisciplinary team composed of pharmacists, experts in infection control, anaesthesiologists/intensivists, microbiologists, risk management specialists and surgeons, coordinated by infectivologists with expertise in antimicrobial stewardship.

The study included elective surgical procedures performed in 2018 in our hospital in 12 main surgical units (digestive surgery, endocrine surgery, general surgery, hepato-pancreato-biliary surgery, thoracic surgery, urologic surgery, cardiac surgery, vascular surgery, breast surgery, neurosurgery, orthopaedic surgery and ear-nose-throat surgery).

In our hospital the patients’ data were reported using an electronic health chart while the information about the patients in the operating room were recorded using healthcare software (https://www.dedalus.eu).

The study was conducted in three sequential phases: phase 0, phase 1 and phase 2.

Phase 0. In April 2017, the AMS team together with surgeons, considering the international and national guidelines, defined the Terni hospital evidence-based guidelines for the optimal use of SAP, also synthesized in a pocket card.

To improve the organization and resolve the logistic problems, a new workflow was defined to optimize the process of ordering, dispensing, administering and documenting the SAP.

A satellite pharmacy was created in the operative block, dispensing only the antibiotic planned for SAP by hospital evidence-based guidelines reported in Table 1.

Every antibiotic box was labelled by a barcode.

Step 1. The surgeon’s prescription is recorded in a computerized clinical chart.

Step 2. The patient arrives in pre-operating room.

Step 3. The nurse takes the antibiotic from the satellite pharmacy and traces the antibiotic by barcode using a healthcare software program.

Step 4. The anaesthesiologist administers the antibiotic.

The workflow is reported in Figure 1.

In December 2017, the Terni hospital evidence-based guidelines for the optimal use of SAP were published on the hospital’s website [19].

Educational and informative events about the Terni hospital guidelines and the new workflow were organized.

Phase 1. Two authors (B.T., G.P.) analysed 2059 elective surgical procedures in the 12 surgical wards from January to June 2018, looking for three SAP appropriateness parameters: indication, choice and dose. The data are reported in Table 2. Phase 2. In July 2018, an audit involving surgeons, nurses and anaesthesiologists was organized, concerning SAP prescription/administration and results from January to June 2018. After review and evaluation of the data and a thorough discussion, recommendations for improvement were issued.

Successively, the same two authors analysed 1781 elective surgical procedures from July to December 2018 for the same three SAP appropriateness parameters: indication, choice and dose. The data are reported in Table 2.

## 3. Statistical Analysis

To test the effectiveness of the AMS intervention’s educational audit, percentages of compliance with the recommendations in phase 1 and phase 2 were compared through the standard Pearson’s Chi-squared test (software R-basic package v. 3.3.2), with a significance threshold of α = 0.05.

## 4. Results

In phase 1, from January to June 2018, 2059 elective surgical, consecutive procedures were undertaken in 12 surgical wards.

The indication of antibiotic administration was in most cases appropriate, with a rate of 73.6% (1516/2059).

The indication was inappropriate in 26.4% (543/2059) of cases as patients (98%) received a prophylactic antibiotic, and it was also inappropriate if the American Society of Anaesthesiologists’ physical status classification (ASA score) was <3, because the indication to SAP required an ASA score ≥ 3.

In 1516 elective surgical consecutive procedures the appropriate indication to SAP was evaluated. The choice of antibiotic was appropriate with a rate of 78.4% (1188/1516). The most frequent inappropriate choice of antibiotic was the utilization of cefazoline (first-generation cephalosporin) instead of cefuroxime (second-generation cephalosporin).

In 1188 surgical procedures with an appropriate choice of antibiotic, the compliance to dose was evaluated. The dose was appropriate with a rate of 69.7% (828/1188). In all cases, the inappropriate dose of antibiotic was the administration of 1 g of cefazolin instead of 2 g.

The overall compliance for the 3 appropriateness SAP parameters evaluated was 40.2% (828/2,059).

In phase 2, after the audit, from July to December 2018, 1781 elective surgical consecutive procedures were evaluated for compliance to the same three parameters of SAP appropriateness: indication, choice and dose.

The indication to SAP was appropriate with a rate of 77.8% (1386/1781). The compliance with appropriate choice of antibiotic was correct in 78.4% (1087/1386) of cases. In 911 of 1087 (83.8%) cases the compliance to dose was appropriate. The overall compliance for the three SAP appropriateness parameters was 911/1781 (a rate of 51.1%).

In both phase 1 and phase 2, only antibiotics included in the guidelines were used. The most frequently used antibiotic was cefazolin (90% of cases).

The comparative analysis between phases 1 and 2 indicates that the correct indication had a significant improvement (*p*-value 0.00128), moving from 73.6% (correct indication in 1516/2059 patients) in phase 1 to 77.8% (correct indication in 1386/1781 patients) in phase 2. The choice of antibiotic did not show any significant improvement (*p*-value 0.4863), remaining almost the same: 78.4% (1188/1516) in phase 1 versus 78.4% (1087/1386) in phase 2. The correct dosage significantly improved (*p*-value < 2.2 × 10^−16^), moving from 69.7% (828/1188) in phase 1 to 83.8% (911/1087) in phase 2. The overall compliance had a significant improvement (*p*-value < 5.6 10^−12^) passing from 40.2% (828/2059) in phase 1 to 51.1% (911/1781) in phase 2 (Figure 2).

## 5. Discussion

This is the first prospective study in which not only the compliance of surgeons to local guidelines for antibiotic prophylaxis was evaluated, but also the effectiveness of our AMS intervention to increase the compliance to SAP in Italy.

Although national and local guidelines are developed and published more frequently, the adherence in SAP has remained very low.

Most of the studies published in the literature are observational and retrospective, and they analyse the compliance to SAP by reviewing medical charts. Only a few authors have reported some suggested actions from the European Centre for Disease Prevention and Control (ECDC) to increase compliance:-“Shifting the responsibility of SAP administration to the anaesthesiologist”.-“Audit and feedback”.-“Education/training”.-“Implementation of standardized order form”.-“Implementation of a multidisciplinary management team”.

One of the main actions of the ECDC, reported from Rodríguez-Caravaca et al. [20], was the change of preoperative antibiotic administration from the surgeons to the anaesthesiologists and perioperative nursing staff. The objectives of this change are represented by the achievement of a proper timing and clearly defined roles/responsibilities for antibiotic administration. In effect, the early or late antibiotic administration has always represented the Achilles’ heel of SAP. This problem is often consequent to an unclear role of surgical and anaesthetics staff in the operating theatre. Physicians and nurses do not have a lack of awareness about the need of the antibiotic prophylaxis. However, there are some communication problems in NOTSS (non-technical skills for surgeons) and between the different components of the operating team.

Audits and feedback are the most common ECDC actions reported in the literature. In fact, modern healthcare systems always track performance outcomes to look for ways to improve quality of care.

The educational program is usually performed through meeting with hospital surgeons and lectures about Surgical Antibiotic Prophylaxis (indication, type, timing of administration, dose and duration).

The implementation of standardized order forms has been reported only from Garcia et al. [21], suggesting the use of a specific prophylaxis kit.

In this study we performed all the five actions suggested by the ECDC to increase the compliance in SAP. At present, the prospective realization and the complete adherence to ECDC guidelines may be the most important results.

This study was designed by our hospital AMS team. It attempted to overcome the obstacles preventing the implementation of adequate SAP by the definition of hospital guidelines on SAP with the involvement of surgeons as well as the creation of a protocol which has standardized the workflow and tried to overcome the organizational and logistical problems.

In our protocol, we have programmed all the ECDC actions:The responsibility of SAP administration was shifted to the anaesthesiologists;Audits with feedback on appropriateness of prescription were performed every six months;Medical grand rounds were performed to promote the excellence and quality in Surgical Antibiotic Prophylaxis;The synergism with pharmacists permitted the organization of an operative block satellite pharmacy for antibiotic prophylaxis; andAll the actions were performed by a multidisciplinary team.

In our study, the application of the five ECDC actions, the implementation of a model with a definition of the workflow and accountability and the systematic collection of data by a surveillance team granted a meaningful implementation of the compliance with a significant improvement of appropriate indication (*p*-value 0.00128), appropriate dosage (*p*-value < 2.2 × 10^−16^) and overall compliance (*p*-value <5.6 × 10^−12^) between phases 1 and 2.

The satellite pharmacy provided only eight types of antibiotics (cefazolin, clindamycin, vancomycin, gentamicin, metronidazole, cefuroxime, cefoxitin and doxycycline), all quoted in local hospital guidelines, without the possibility to prescribe other antibiotics, which led to a very high utilization of cefazolin (90%). However, there was no significant improvement in the choice of antibiotic (*p*-value 0.4863), therefore it will be necessary to organize more audits concerning the appropriateness of the utilization of first- and second-generation cephalosporins.

We believe that more studies should analyse the effectiveness of the intervention instead of evaluating only the compliance.

Our study has a limitation: we could not analyse the timing and the possible redosing of SAP. Several studies have confirmed the relationship between the timing of SAP and SSI risk and the importance of maintaining adequate antibiotic serum concentrations throughout the operation [21,22,23,24,25,26,27]. The monitoring of these two important parameters will also be included in our analysis in the near future.

## 6. Conclusions

The appropriateness of SAP is a key component to reduce SSIs [28]. Approximately 15% of all antibiotics in hospital are prescribed for SAP [29], and this can be a major cause of some emerging infections [30,31] and of the selection of antibiotic resistance, increasing healthcare costs. Many guidelines have been published, but the overall compliance remains poor. There is no universally recognized intervention to improve the appropriateness of SAP. In this study we suggest an antimicrobial stewardship intervention that seemed to improve appropriateness.

## Figures and Tables

**Figure 1 antibiotics-09-00168-f001:**
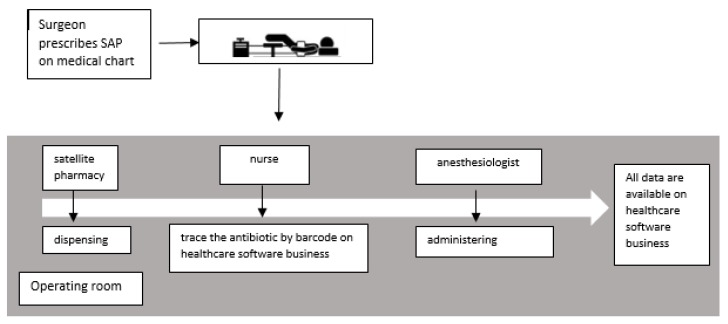
Workflow of ordering, dispensing, administering and documenting, defined by Terni hospital evidence-based guidelines for the optimal use of SAP.

**Figure 2 antibiotics-09-00168-f002:**
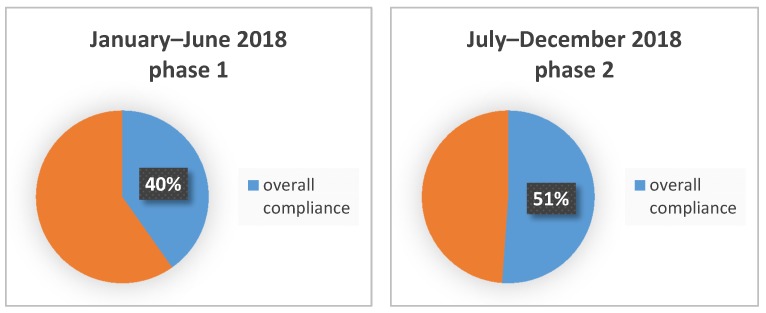
Overall compliance to SAP in phase 1 (January–June 2018) and phase 2 (July–December 2018).

**Table 1 antibiotics-09-00168-t001:** Dose and half-life of antibiotics planned for Surgical Antibiotic Prophylaxis (SAP).

Drug	Dose (Adult)	Half-Life
Cefoxitin	>50 kg 2 g IV <50 kg 1 g IV	1 h
Cefazolin	>50 kg 2 g IV <50 kg 1 g IV	1.2–2.2 h
Cefuroxime	2 g IV	1 h
Clindamycin	600 mg IV	3 h
Gentamicin	3 mg/kg IV	1–2 h
Vancomycin	15 mg/kg IV Max dose 1 G	4–6 h
Doxycycline	200 mg os before surgery (induced abortion)	6–12 h
Metronidazole	500 mg IV	8 h

**Table 2 antibiotics-09-00168-t002:** Results of elective surgical procedures for 3 SAP parameters of appropriateness: indication, choice, dose.

	January–June 2018	July–December 2018	*p*-Value
Elective surgical procedures	2059	1781	
Appropriate indication	73.6% (1516/2059)	77.8% (1386/1781)	0.00128
Appropriate antibiotic choice	78.4% (1188/1516)	78.4% (1087/1386)	0.4863
Appropriate dose	69.7% (828/1188)	83.8% (911/1087)	<2.2 × 10^−16^
Overall compliance	40.2% (828/2059)	51.1% (911/1781)	<5.6 × 10^−12^

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
