# Peer review of "Impact of Antimicrobial Stewardship Interventions on Appropriateness of Surgical Antibiotic Prophylaxis: How to Improve"

_antibiotics, 2020, doi:10.3390/antibiotics9040168_

Round 1

Reviewer 1 Report

Reviewed article demonstrated a model of successfully antimicrobial 55
stewardship intervention that improves appropriateness on SAP.

Materials and methodology used by the authors are appropriately selected. The study group is also representative. In the analysis carried out, the effectiveness of the created system was proved. There was an increase in the correct indication (from 73.6 to 77.8% in the first and second phase respectively), a very large increase in the correctness of the prescribed dose (from 69.7% to 83.3% in the first and second stages respectively) and a significant increase in overall compliance (from 40.2% to 51.1% in the first and second phases respectively).

Editorial comments:

  • line 98: should be patient instead of patient's
  • line 127: the link to the page will look better as a reference in the bibliography
  • lines 142-144: looks they are unnecessary spaces
  • line 149: missing space (α =0.05 instead of α = 0.05)
  • lines 170-176: should be uniform, without enter
  • line 201: the reference name should be a regular font, not italic
  • line 214: not "et coll", should be "et al."
  • line 234: in "(p-value< 2.2 10-16)", -16 as upper index
  • line 235: in "<5.6 10-12", -12 as upper index
  • line 258: should be We instead of we

In my opinion, the article meets all the requirements to publish it after making the proposed minor corrections.

Author Response

Dear reviewer thanks you a lot for your revision. We did all the corrections required.

Reviewer 2 Report

The main theme of the manuscript of Tiri et al. is the description of a prospective study that demonstrated the success of a model of antimicrobial stewardship intervention that improves appropriateness of Surgical Antibiotic Prophylaxis however, it is necessary to clarify some aspects as suggested in order to improve the manuscript.

In conclusion, a major and substantial revision is suggested.

Comments:

The title clearly and precisely reflects the finding of the manuscript.

 The Abstract is clear and is in line with the context however:

The abstract is clear, but the sentence lines 52-54 should be completed.

Introduction

The introduction should be improved:

Please introduce more comments regarding the antimicrobial resistance, the role of opportunistic pathogens in hospital acquired infections particularly referred to the role of biofilm.

Please add more information about the antimicrobial resistance phenomenon in Italy and in your region

Results and Discussion

The results are difficult to interpret and appear as a list of percentages. The authors should represent the data obtained through pie charts.

The authors should indicate the list of the antimicrobials used as well as the corresponding dosage highlighting the most effective ones.

what type of surgery have the patients examined undergone?

do the authors know how many patients developed infection and the microorganisms responsible of infections in patients in whom prophylaxis was unsuccessful?

moreover, the authors should indicate the guidelines, to provide a wider overview to the   reader

References

Please check the references accurately, according to the journal guidelines

Author Response

Dear reviewer thanks you a lot for your revision and yours contribute to improve the article. We did all the corrections required and we reviewed the work in light of his valuable suggestions.

Round 2

Reviewer 2 Report

The authors replied to almost everything therefore the paper could be suitable for publication in Antibiotics.

Author Response

Thank you for your reply.